# Quantifying the relative importance of genetics and environment on the comorbidity between mental and cardiometabolic disorders using 17 million Scandinavians

Joeri Meijsen [1] ✉, Kejia Hu [2], Morten D. Krebs [1], Georgios Athanasiadis [1,3], Sarah Washbrook [4], Richard Zetterberg [1], Raquel Nogueira Avelar e Silva [1], John R. Shorter [1,5], Jesper R. Gådin [1], Jacob Bergstedt [2], David M. Howard [6], Weimin Ye [7], Yi Lu [7], Unnur A. Valdimarsdóttir [7,8,9], Andrés Ingason[1], Dorte Helenius [1], Oleguer Plana-Ripoll [10], John J. McGrath [11,12,13], Nadia Micali [1,4,14], Ole A. Andreassen [15,16,17], Thomas M. Werge [1,18], Fang Fang [2] & Alfonso Buil [1] ✉

Mental disorders are leading causes of disability and premature death worldwide, partly due to high comorbidity with cardiometabolic disorders. Reasons for this comorbidity are still poorly understood. We leverage nation-wide health records and near-complete genealogies of Denmark and Sweden (n = 17 million) to reveal the genetic and environmental contributions underlying the observed comorbidity between six mental disorders and 15 cardiometabolic disorders. Genetic factors contributed about 50% to the comorbidity of schizophrenia, affective disorders, and autism spectrum disorder with cardiometabolic disorders, whereas the comorbidity of attention-deficit/hyperactivity disorder and anorexia with cardiometabolic disorders was mainly or fully driven by environmental factors. In this work we provide causal insight to guide clinical and scientific initiatives directed at achieving mechanistic understanding as well as preventing and alleviating the consequences of these disorders.

Individuals with mental disorders (MD) have substantially higher rates of various medical conditions[1] and mortality[2] as compared to the general population. Indeed, individuals with a mental disorder diagnosis face a 7- to 10-year shorter life expectancy[3]. This is mainly due to the 2- to 3-fold increased risk of premature death from somatic comorbidities such as cardiometabolic disorders (CMD)[3–5], including coronary artery disease (CAD)[6], stroke[7], and heart failure (HF)[8]. These disorders are often preceded by other related conditions such as type-

2 diabetes (T2D), high cholesterol, obesity, and hypertension. Later epidemiological studies[1,9,10] systematically leveraging population-wide, healthcare registers confirmed and extended the positive MD-CMD association and found it to be bidirectional, i.e., also mentally healthy individuals at the time of CMD onset, go on to develop mental illness more frequently than expected by chance.

Importantly, risk increases may vary considerably across pairs of individual mental and cardiometabolic diagnoses. In fact, individuals

**Fig. 1 | Cumulative incidences and 95% confidence intervals of ADHD for individuals born 1981–2005 using medical records up to 2012 stratified by year of birth.** Cumulative incidences are shown for (**A**) the general population ($n = 1,560,901$ individuals), (**B**) individuals with at least one full-sibling diagnosed with ADHD ($n = 24,476$ individuals), (**C**) individuals with at least one parent diagnosed with type-2 diabetes ($n = 5557$ individuals). Confidence intervals are reported as cumulative incidence estimates at a given age for a specific year of birth +/− 1.96 × standard error.

diagnosed with anorexia nervosa (AN), attention-deficit/hyperactivity disorder (ADHD), or an affective disorder (AFF) are more likely to be diagnosed with atrial fibrillation later in life than individuals with schizophrenia (SCZ) or autism spectrum disorder (ASD), whose risk may be lower than the background population[1]. The observations of complex patterns of comorbidities indicate considerable diversity in the underlying causes, which may include both shared genetic architectures and environmental exposures such as prescription drugs, substance abuse, socioeconomic conditions, and traumatic live events.

An important step toward designing effective prevention and treatment strategies to combat these comorbid conditions would be effectively discriminating heritable genetic effects and environmental risk exposures for each comorbid pair of MD and CMD. When the comorbidity is driven mainly by environmental factors, clinical approaches will prioritise interventions in the environment. However, when genetics plays an important role, therapeutic approaches could also benefit from precision medicine tools such as genetic risk prediction models.

Notably, progress is complicated by extensive correlation among, as well as between, environmental risk exposures and inherited disease liability, challenging the identification of causative factors[11]. Previous studies have found a substantial genetic overlap between CMD and MD[12–15], which suggests genetics contribute to the MD-CMD comorbidity. However, these studies obtain their estimates for genetic overlap from summary statistics of genome-wide association studies (GWAS), a strategy that suffers from serious limitations: first, these studies are affected by the extreme sampling of cases suffering from comorbidities other than the index diagnosis under study, and of control participants recruited among overly healthy subjects, and thus, they are genuinely unrepresentative of the background population[16]; second, the current examples deal with specific disorders, and different populations and methods, but a systematic analysis for a large spectrum of disorders in the same population is lacking; and third, the estimates are sensitive to the underlying assumptions of the statistical model and the linkage disequilibrium panel used to calculate them[17]. Moreover, these previous studies report genetic correlations as a measure of shared genetic effects between two disorders but do not investigate the relative importance of genetic and environmental factors on the comorbidity between disorders.

In this work, we leverage nationwide healthcare registers and near-complete population genealogies spanning 4 generations of Denmark and Sweden for 17 million individuals to quantify the genetic contribution underlying the comorbidity between 6 mental disorders

and 15 cardiometabolic disorders. Nationwide genealogies cross-referenced with sociodemographic and comprehensive healthcare registers from an exclusively public healthcare system provide an attractive opportunity to perform large-scale and population-true analyses that can discriminate shared heritable from environmental factors across individuals in extended families. For each country, we first estimate the incidence of 6 mental disorders and 15 cardiometabolic diseases in the general population and for individuals with affected relatives. Next, we determine the heritability for each single disorder as well as the genetic correlations between all pairs of diagnoses. Finally, we quantify the relative contribution of genetic factors for each of the comorbid MD and CMD constellations.

## Results

Disentangling and quantifying the causes of clinically important and aetiologically complex constellations of diseases, such as comorbid mental disorders with cardiometabolic diseases, relies on the ability to reliably observe manifestations of both classes of diseases in large populations of related individuals over decades. However, differences in age-at-onset, length-of-observation periods, population structure, healthcare provision and ascertainment biases have traditionally compromised our understanding of the classical nature-nurture duality of causalities.

Here we circumvent these challenges by leveraging two unique, near-complete population genealogies: those of Denmark and Sweden, including 6 and 11 million individuals across 4 generations, respectively. The two populations have been served for decades by similar, public healthcare systems. Clinical diagnoses of diseases have been systematically and uniformly recorded for 39 and 43 years, and data made available for analyses alongside population-wide information on individual-level, familial relatedness.

### Register heritability ($h^2$) and genetic correlation ($r_g$) estimates

To estimate heritability, genetic correlations, and environmental risk components across the six selected mental disorders and 15 mostly common and severe cardiometabolic disorders, we first determined the cumulative incidence function (CIF) across birth years[18,19]. As exemplified for ADHD (Fig. 1A) and for all six mental disorders (Supplementary data 1A–G), CIFs vary systematically across birth years, and year-based CIFs are therefore included in all subsequent analyses of heritability and genetic correlations as outlined in detail below.

Notably, the CIF for individuals with a sibling with ADHD (Fig. 1B), as well as the CIF for individuals with a parent diagnosed with type-2

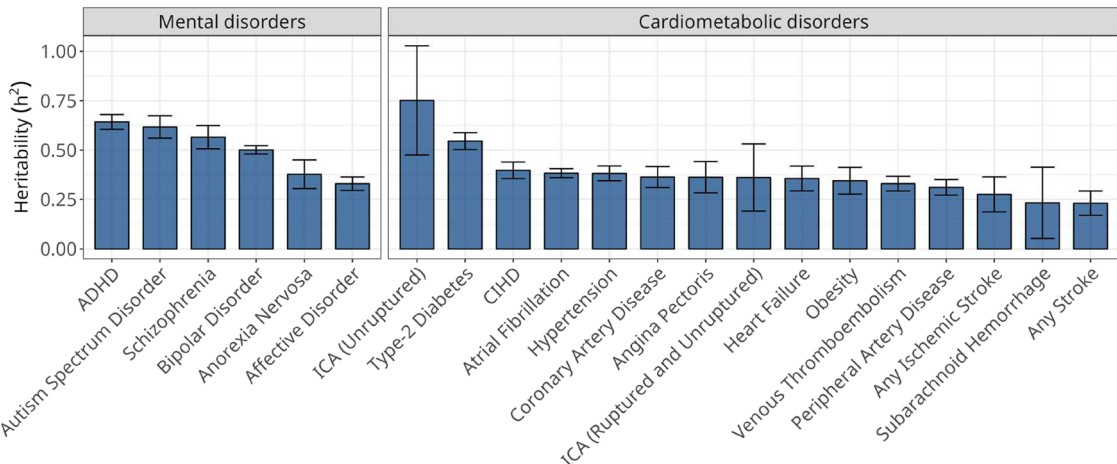

**Fig. 2 | Meta-analysis of Danish and Swedish narrow sense heritability (h²) estimates and 95% confidence intervals of mental- and cardiometabolic disorders.** Estimates were calculated using information from all individuals born in Denmark (n = 7,797,720) and Sweden (n = 13,222,453). ADHD attention deficit/ hyperactivity disorder, ICA intracranial aneurysm, CIHD chronic ischaemic heart disease. Confidence intervals are reported as the meta-analysis of Danish and Swedish narrow sense h² estimates +/− 1.96 × standard error.

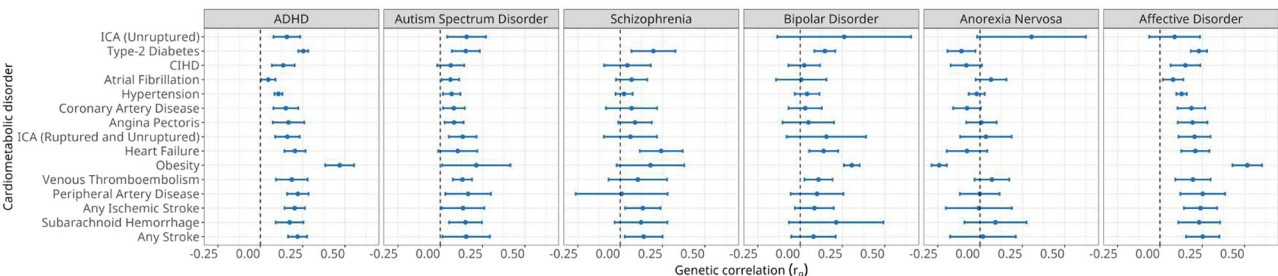

**Fig. 3 | Meta-analysis of Danish (n = 1,560,901) and Swedish (n = 2,566,100) genetic correlations (r_g) estimates between mental- and cardiometabolic disorders and 95% confidence intervals calculated using national register data.** Individuals were born between 1981 and 2005 using medical records up to 2012. ADHD attention-deficit/hyperactivity disorder, ICA intracranial aneurysm, CIHD chronic ischaemic heart disease. Confidence intervals are reported as the meta-analysis of Danish and Swedish r_g estimates +/− 1.96 × standard error.

diabetes (Fig. 1C) are markedly increased relative to the CIF of ADHD for the entire population (Fig. 1A). When comparing the year of birth matched cumulative incidences reported in Fig. 1A, B, we see a substantial difference in ADHD incidence in the general population (Fig. 1A) versus individuals with an affected relative (Fig. 1B). This difference highlights the importance of genetic effects while trying to account for substantial non-genetic components. These observations are consistent with ADHD being highly heritable and with a significant genetic-correlation between ADHD and type-2 diabetes[20]. All Danish CIFs are listed in Supplementary data 1A–G.

We applied the analytical approach reported by Wray and Gottesman[21] to systematically estimate heritability and genetic correlations for all six mental and 15 cardiometabolic disorders in the Danish and Swedish populations. As shown in Fig. 2, heritability estimates range from 25% to 75% and were strongly correlated between the two countries (Figure S1A and Supplementary data 2) and are broadly concordant with existing studies (Supplementary data 3). Similarly, we estimated genetic correlations for each of the 90 diagnostic MD-CMD pairs in both populations and observed 32 significant correlations (Bonferroni $p < 5.95 \times 10^{-4}$) after meta-analysing the highly similar country-specific estimates (Fig. 3 and Figure S1B). With the notable exception of AN, the genetic correlations of the mental disorders with the 15 cardiometabolic disorders were positive. They were larger for ASDs and bipolar disorder (BD) than for ADHD and AFFs (Supplementary data 4A).

It is well known that the comorbidity between two diseases depends not only on their genetic and environmental correlations but also on their heritability[22,23]. Genetic correlation alone—estimating the degree of overlap of the genetic architectures of two disorders—may account for only a negligible fraction of the observed comorbidity. This scenario may occur for disorders with high genetic correlations when the heritability of one or both disorders is modest, as shown in simulated data in Figure S2. To determine the relative contributions of heritable genetic variants and environmental risk factors to MD-CVD comorbidities, we applied quantitative genetic methods that combined the estimates of heritability and genetic correlation for each pair of disorders. As measures of comorbidity, we used hazard ratios previously determined in the Danish population[1], allowing us to estimate the relative heritable and environmental components for 35 of the 90 pairs of MD-CVD. As shown in Fig. 4, the environmental component is comparable and, in most cases, markedly higher than the genetic component of the comorbidity. Notably, the genetic components for the comorbidity between all CMDs with AN was negligible and it was also very low with ASD. For all MDs, the comorbidity with hypertension and atrial fibrillation has a lower genetic component than for other CMDs. As observed for ASD, negative phenotype correlation with CMDs is driven by protective environmental factors. All estimates (i.e., hazard ratios, cardiometabolic prevalence's, r_p, G, E) including SE, z-scores, CIs, and p-values are reported in Supplementary Data 5.

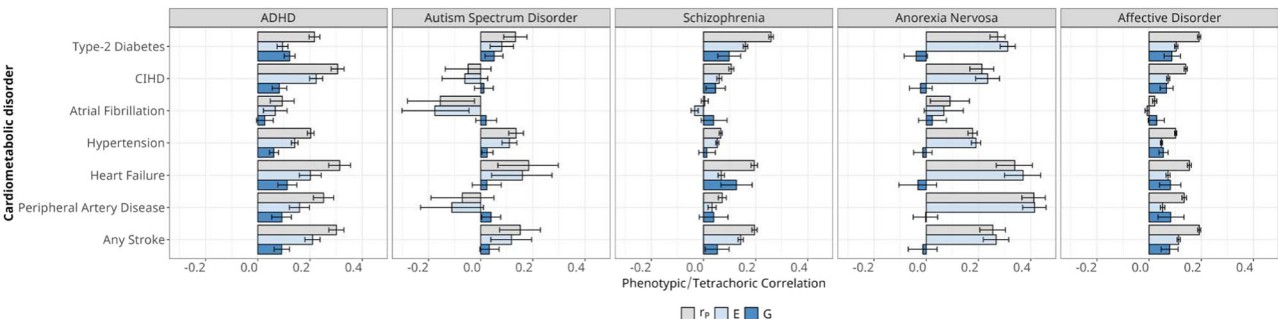

**Fig. 4 | Quantification of the contribution of genetic (G) and the non-genetic (E) factors to the observed phenotypic correlation ($r_p$) between mental- and cardiometabolic disorders using register based genetic correlations ($r_g$) and heritability ($h^2$) estimates.** Estimates of $r_p$ were selected from Momen et al. [1]. Individuals were born between 1981 and 2005 using medical records up to 2012 ($n = 1,560,901$). Confidence intervals are reported as either (a) phenotypic correlation estimate +/− 1.96 × standard error, (b) non-genetic (E) component +/− 1.96 × standard error, or (c) genetic (F) component +/− 1.96 × standard error.

In summary, the observed MD-CMD comorbidity patterns are mostly explained by shared environmental rather than heritable factors, a trend that is particularly pronounced for AN and ASD, as well as for hypertension and atrial fibrillation.

## Comparison using genotype information

Since most studies on heritability and genetic correlations are performed using GWAS summary statistics, we compared our results with those obtained using single nucleotide polymorphism (SNP)-based heritabilities ($h^2_{SNP}$) and genetic correlations ($r_{g\ SNP}$). We used LDSC and summary statistics from well-powered GWAS (Supplementary data 6). To make results comparable with our register-based results we used MD GWAS for the large Danish cohort iPSYCH 2012[24,25]. We converted these estimates to a liability scale[26].

On a liability scale, the median of $h^2_{SNP}$ across all MD combined was 0.17 (SD = 0.03), lowest for BD (0.13; 95% CI 0.03−0.24) and highest for ADHD (0.23; 95% CI 0.2−0.27) for ADHD. The $h^2_{SNP}$ across CMD was lower with a median of 0.09 (SD = 0.06), lowest for heart failure ($2.3 \times 10^{-2}$; 95% CI = $1.93 \times 10^{-2} - 27 \times 10^{-2}$) and highest for unruptured ICA (0.22; 95% CI 0.13−0.32). Thus, relative to the register-based heritability estimates ($h^2$), the $h^2_{SNP}$ were lower for MD (63%) and CMD (72%). Correspondingly, $h^2_{SNP}$ for the individual MD and CMD disorders), were significantly smaller than the register-based $h^2$ estimates, except for SAH and ICA (Figure S3 and Supplementary data 7).

Regarding SNP-based genetic correlations, we observed 21 substantial and significant genetic correlations, of which 12 were associated with ADHD, 6 with AFFs, and the remaining three with AN, ASD, and SCZ (Bonferroni multiple testing correction threshold $p < 5.95 \times 10^{-4}$). These SNP-based genetic correlations were not significantly different from the register-based Danish estimates (Figure S4, Bonferroni $p < 5.95 \times 10^{-4}$), nor from Danish and Swedish meta-analysed genetic correlations (Supplementary data 8). To test if these results are dependent on the specific GWAS used, we repeated the analyses using the larger PGC meta-analysis summary statistics and observed no significant differences compared to the use of the Danish-only GWAS summary statistics (Figure S5).

The high concordance between the register- and the SNP-based genetic correlation estimates confirms a substantial shared genetic contribution, based on common genetic variants between many MDs and CMDs. However, since the genetic contribution to comorbidity also depends on the heritability of both diseases, the comorbidity partition using SNP-based estimates largely underestimates the genetic component (Figure S6). This fact highlights the importance of family-based estimates derived from large national registers.

## Discussion

One of the main challenges in healthcare is to understand the causes of complex patterns of comorbidity. Here we provide the first systematic and comprehensive analyses to discriminate and quantify environmental and heritable causes underlying comorbidity between the clinically important disease domains of mental and cardiometabolic disorders. Leveraging unique genealogies and cross-referenced, nationwide healthcare registers holding near-complete records from the public healthcare systems in Denmark and Sweden, we document that the notable comorbid manifestations of single pairs of mental and cardiometabolic disorders are more environmental (50−70%) than heritable in origin. Importantly, these findings show that the nature-nurture origin of comorbid presentations cannot be deduced from estimates of the genetic correlation of the corresponding disorders. Moreover, we show that the relative contribution of genetic and environmental causes to the observed comorbidities between MD and CMD varies largely and is specific for each disease pair.

We analysed six mental disorders and 15 cardiometabolic diseases, and we observed large variability between the relative contribution of genetics and environment to the comorbidity of the 90 disease constellations. We observed that the observed comorbidity between anorexia nervosa and most CMDs can be explained exclusively by non-genetic effects, while the genetic component plays a substantial role (around 50%) in the comorbid presentations of affective disorder with most of the examined CMDs. We calculated the $r_g$ between MDD and all CMD as observed no significant difference with the AFF results supporting out assumption that MDD makes up the majority of AFF diagnoses (Supplementary data 4, B). Constellations involving CMDs with ADHD or schizophrenia show an intermediate genetic contribution of around 30%. In summary, the environmental and genetic causes of comorbidity vary considerably across pairs of disorders and cannot be deduced simply from the heritability of the individual disorders (or their genetic correlation); rather insight into the origin of comorbidity requires individual analyses and interpretation. This general principle is illustrated by two pairs of diagnoses, that of AFF and hypertension, and between ASD and T2D. Both pairs are characterised by a low genetic correlation of 0.1; however, in the first pair, 50% of the comorbidity can be explained by genetic factors, whereas in the second pair, the genetic contribution is not different from zero. This shows that the same genetic correlation estimate could result in different estimates for the relative importance of genetics and environment on the comorbidity between two disorders. Therefore, to appreciate both the quantitative and qualitative composite nature of the causality underlying comorbid presentations, each disease pair should be considered individually.

The interpretation of these results requires some discussion on the nature of environmental effects. There are two diverse types of

environments, one related to the behaviour of the individual itself while the other is mediated by external agents. The environments created by the individual behaviour may stem from the mental disorders themselves, e.g., physical immobility, irregular sleep, and overeating are characteristic of many MDs, while compulsive behaviour and derived alcohol misuse, smoking, and overeating are specifically linked to ADHD. External sources of environmental risk in mental disorders include second-generation antipsychotics, most notably clozapine, prescribed to individuals with SCZ. These medications provoke overeating, malnutrition, obesity, long-QT, dyslipidaemia, and electrolyte deregulation, all of which are well-documented risk factors for CMD[27]. As the most extreme example, we provide evidence that the comorbidity between AN and T2D can be explained exclusively by environmental factors. In this case, the combination of an individual's behaviour and external environmental factors associated with the treatment for AN could be responsible for the AN-T2D comorbidity. For example, dietary restriction and malnutrition in individuals with AN cause long-term effects on endothelial tissues, increasing the risk of T2D[28]. Furthermore, individuals with active AN are subject to a metabolic system that is no longer working at normal capacity, which increases the risk for T2D when AN is being treated using a high-calorie diet during inpatient refeeding. An additional environmental challenge in individuals with AN is smoking, which has been shown to be more prevalent due to its appetite suppressant qualities[29].

Our results offer a much-needed discussion of the mechanisms of comorbidities and suggest further lines of investigation into the causes of MD-CMD comorbidity. The observation of a substantial genetic component in some MD-CMD comorbidity constellations justifies attempts to identify the genetic determinants of comorbidity between specific MD-CMD pairs that may have clinical implications. First, studies aiming to discriminate genetic underpinnings of single disorders from the genetic determinants of comorbid conditions would inform us about pleiotropic disease processes across different organ systems, as well as guide drug discovery or repurposing. Our results also highlight the importance of environmental factors as major causes for comorbidity between many MD-CMD pairs. However, new studies are needed to identify the relevant environmental exposures for each MD-CMD pair.

A strength of this study is the use of nationwide registers in Denmark and Sweden. These countries share many similarities in history, (genetic) ancestry, cultural characteristics, and their government-funded healthcare system. However, clinical variations such as different recording of diagnosis periods, use of different ICD versions[30], and varying diagnostic practices have resulted in country-specific time trends. Despite this, replication analysis of heritability and genetic correlation estimates derived using the larger Swedish registers showed no significant difference from the Danish estimates. The high transferability between the two genealogies adds validity to the findings on heritability and genetic correlations. On the other hand, the explosion of published GWAS in the last years has been followed by an explosion of papers estimating the heritability and genetic correlations between diseases based on GWAS summary stats. Even though it is known and accepted that the SNP-based estimates are largely underestimates of the real values, SNP-based estimates are becoming the standard. Here we took advantage of the unique opportunity to compare SNP-based and register-based estimates of heritability and genetic correlations using almost the same set of individuals within a whole country. To our knowledge, this is the first time that such a comparison has been done in a large sample. Our results confirm the fact that SNP-based heritability estimates capture only a small fraction of the total heritability and reveal that this fact has important consequences when decomposing genetic and environmental factors between two diseases.

To complement our register analysis, we computed SNP-based estimates of heritabilities and genetic correlations leveraging molecular genetic data on the Danish population-based sample (iPSYCH 2012 cohort). To our knowledge, this is the first comparison of heritabilities and genetic correlations based on genotypes and register data carried out in the same population.

Our study has also some limitations. First, the study leverages secondary care hospital diagnoses from 1968 onwards, and due to truncation, some diagnoses introduced more recently are underrepresented in older individuals. However, our strategy of estimating heritability and genetic correlations by by-year meta-analyses partially controls for these biases in the data. Second, we did not quantify the influence of shared environment in close relatives on the heritability and genetic correlation estimates, and therefore these estimates might be slightly inflated, although previous studies suggest that such effects are small[21]. Third, in our model, we assume no interaction between the genetic and non-genetic factors that contribute to the comorbidity between mental and cardiometabolic disorders, which may affect the estimated effect sizes. Fourth, the use of hospital records may represent a more extreme phenotype as compared to medical records from family doctors and general practitioners, however, we hypothesise that this is both country and disorder-specific. Finally, while the Danish sample size is large, and results replicated well in the Swedish register data, it is uncertain whether our results translate to non-Scandinavian countries and populations using different healthcare systems inside and outside Europe. However, we think it is reasonable to expect that countries with similar universal health systems and levels of development would give similar results[30].

To the best of our knowledge, this study is the first to quantify the relative contribution of genetic and environmental factors to the comorbidity of different constellations of mental and cardiometabolic diseases by integrating health registers and genealogical information from two countries and cross-referencing with molecular genetic data from a representative population sample. This work provides a foundation to guide future precision medicine by helping to implement clinical interventions to prevent and treat CMD in individuals with mental disorders.

## Methods
### Integrative psychiatric research consortium 2012 (iPSYCH2012) cohort
The iPSYCH2012 cohort is a well-documented extensively cited cohort[24]. In short, individuals born between 1981 and 2005 ($n = 1,472,762$) were considered for ascertainment, representing the entire population of Denmark born in that timeframe. Of these 30,000 were randomly sampled, regardless of (psychiatric) disorder status, to create an unbiased population representative control group. Using information ascertained from the Danish Civil[31,32], National Patient[33] and/or Psychiatric Central Research Registers[34] 57,764 design cases were selected with indications of clinical diagnoses of mental health disorders. In total, 87,764 individuals were selected to form the cohort. Indications are based on International Classification of Disease (ICD) codes representing the clinical diagnosis associated with an instance of care provided at one of many psychiatric facilities throughout Denmark. The following six case groups as having at least one indication with the corresponding ICD 10[35] (or equivalent ICD 8[36]) codes were defined: attention deficit hyperactivity disorder (ADHD: F90.0), anorexia nervosa (AN: F50.0, F50.1), autism spectrum disorder (ASD: F84.0, F84.1, F84.5, F84.8, F84.9), affective disorder (AFF: F30–F39), bipolar disorder (BD: F30-31), and schizophrenia (SCZ: F20). Of all selected individuals a dried neonatal heel prick blood spot was obtained from the Danish Neonatal Screening Biobank[37]. Individuals were removed when no blood spots could be obtained. The use of this data is according to the guidelines provided by the Danish Scientific Ethics Committee, the Danish Health Data Authority, the Danish Data

Protection Agency, and the Danish Neonatal Screening Biobank Steering Committee. For each dried bloodspot the DNA was extracted and amplified followed by genotyping using the Infinium PsychChip v1.0 array[24]. Of 9714 bloodspots not DNA could successfully be genotyped and therefore the individuals were excluded from the study. A subset of good quality SNPs were phased into haplotypes using SHAPEIT3[38] and imputed using Impute2[39] with reference haplotypes from the 1000 genomes project phase 3[40]. Genotypes were checked for imputation quality (INFO > 0.2), Hardy-Weinberg equilibrium (HWE; $p < 1 \times 10^{-6}$), association with genotyping wave ($p < 5 \times 10^{-8}$), association with imputation batch ($p < 5 \times 10^{-8}$), differing imputation quality between subjects with and without psychiatric diagnoses ($p < 1 \times 10^{-6}$), and minor allele frequency (MAF > 0.01). Finally, we extracted unrelated individuals of European ancestry leaving 77,082 individuals.

## Danish nationwide registers

The *Danish Civil Registration System* was established in 1986 and contains detailed information pertaining to sex, date of birth, parental links, and continuously updated information on vital status (e.g., migration or death) for all individuals alive and living in Denmark for the past seventy years[31,32]. The *Danish National Patient Register* includes the full medical records of all individuals treated at Danish hospitals (inpatient department) since January 1, 1977, as well as in outpatient clinics since 1 January 1994 (or occasionally since 1995)[33]. The register was updated in 2002 to also include individuals treated in hospitals outside of Denmark and treatments not covered under the Danish health insurance agreement at private healthcare facilities. Finally, the *Danish Psychiatric Central Research Register* contains data on admissions to psychiatric inpatient facilities up to and including 1994. Following 1994, the register was extended to include outpatient contacts in psychiatric departments[34]. As of April 2017, the civil register contained 9,851,330 individuals, the national patient registers 8,065,597 individuals, and the psychiatric register 1,005,068 individuals. All individuals were born between January 1, 1858, and April 21, 2017. All registers contained a unique personal identification number given to all individuals living in Denmark, therefore allowing for accurate linking across the different registers. By Danish law, informed consent is not required for register-based studies and no compensation was provided. This work is based on Danish register data that are not publicly available due to privacy protection, including the General Data Protection Regulation (GDRP). Only Danish research environments are granted authorisation. Foreign researchers can, however, get access to data under Danish research environment authorisation. Further information on data access can be found at https://www.dst.dk/en/TilSalg/Forskningsservice or by contacting the senior corresponding authors.

## Swedish nationwide registers

The *Swedish Total Population Register (TPR)*, started in 1968 and continuously updated, holds information on all individuals who are residents of Sweden. It contains information on birth, death, name change, marital status, family relationships and migration within Sweden as well as to and from other countries[41]. *Multi-Generation Register (MGR)*[42] is part of TPR and contains information on all residents in Sweden who were born in 1932 or later and alive in 1961 ("index persons"), together with their parents. Familial linkage (i.e., parental information) is available for more than 95% of individuals who died before 1968, about 60% of those died between 1968 and 1990, and more than 90% of those alive in 1991. *The Swedish Inpatient Register* was launched in 1964 (psychiatric diagnoses from 1973) but complete coverage was reached in 1987. It includes discharge diagnoses, dates of hospital admission and discharge, and has a coverage of at least 71% of all residents for somatic care discharge in 1982 and 86% of all psychiatric care in 1973. Since 2001, this register also covers outpatient[43]. The individually

unique National Registration Number was used to link data from all the registers. All Swedish-born residents were followed for any cardiometabolic and mental disorders from birth until emigration or death from 1973 to 2016. By Swedish law, informed consent is not required for register-based studies and no compensation was provided. The use of Swedish data was approved by the regional ethics review board in Stockholm, Sweden with DNR 2012/1814-31/4. Data from Swedish registers are not available for sharing due to policies and regulations in Sweden. Swedish register data are available to all researchers through applications at Statistics Sweden (SCB, https://www.scb.se/en/) and The National Board of Health and Welfare (Socialstyrelsen, https://www.socialstyrelsen.se/)

By Danish and Swedish law, consent to use register data for register-based studies is not required.

## Case definition

We defined the six MDs, namely attention-deficit/hyperactivity disorder (ADHD), anorexia nervosa (AN), autism spectrum disorders (ASD), affective disorders (AFF), bipolar disorder (BD), and schizophrenia (SCZ), and cardiometabolic disorders, using information from the Danish and Swedish Patient Register. Mental disorders were previously defined and used for GWAS analysis of iPSYCH 2012 data by Schork et al. 2019. These disorders represent the most well-documented, well-known and most common mental disorders occurring in the population. The cardiometabolic disorders were selected based on a.) common in the population i.e., high prevalence or b.) less common prevalence i.e., low prevalence and c.) selected disorder had available GWAS summary statistics in any publicly available repository. Note, that AFF includes two main diagnosis, BD and major depressive disorder. Individuals with at least one hospital visit concerning these disorders (primary or secondary diagnosis) were considered cases with MD or CMD. Individuals diagnosed with SCZ, BD, or AFF before age 10 were removed from the analysis, as the validity of such a diagnosis is considered clinically unreliable. ICD 8 codes were used until 1993 and ICD 10 codes were used since 1994 in Denmark; ICD 8 codes were used until 1986, ICD 9 codes were used during 1987–1996, and ICD 10 codes were used since 1997 in Sweden (Supplementary data 9). To minimise the effect of left-handed censuring we removed individuals born outside of Denmark and Sweden as these individuals may have been diagnosed in another country. By doing so we excluded both Danish/Swedish citizens as well as individuals migrating to Denmark and Sweden. No information is recorded regarding terms such as "race", ancestry, or "ethnicity". However, both Denmark and Sweden are predominantly of white-European ancestry with relatively recent large migration patterns from non-European countries therefore we assume that we extracted mostly individuals of white-European ancestry and indirectly removed individuals of non-European ancestry when filtering on country of birth.

## GWAS summary statistics

A total of 15 cardiometabolic GWAS summary statistics including stroke (subtypes)[44], CAD[45], aneurysms[46] and HF[47] were obtained through multiple public repositories. GWAS summary statistics containing participants of the VA Million Veterans Programme (e.g., T2D[48], venous thromboembolism[49], and peripheral artery disease[50]) were provided after approval was granted by the National Institute of Health (*project #26508*). GWAS summary statistics for ADHD[51], AN[52], ASD[53], BD[54], and MDD[55] excluding iPSYCH participants (except SCZ[56] which does not contain iPSYCH samples) were kindly provided through their respective PGC consortium. iPSYCH only GWAS summary statistics for MDs[25] were downloaded from internal iPSYCH servers and are available on request. The full list of all cardiometabolic- and mental disorder GWAS summary statistics used is shown in Supplementary Data 6.

## GWAS summary statistics cleaning

All GWAS summary statistics were uniformly cleaned using internal software[57]. First, for each GWAS summary statistic, we inferred the genome build by mapping SNPs to dbSNP build 151 using GRCh38, GRCHh35, GRCh36, and GRCh37 genomic coordinates. The version with the highest number of mapped SNPs was inferred as the build of the original GWAS. Next, a second mapping step uses the inferred build to simultaneously map and liftover the position and chromosome coordinate to the GRCh37 version of dbSNP, which adds information about reference and alternative alleles. RSids were used when chromosome and base pair information were not available. The reference allele of dbSNP corresponds to the reference allele of the reference genome. The allele directions were flipped making the effect allele the reference allele. Effect scores (e.g., beta coefficients, odds ratios, and z-scores) were adjusted accordingly. Finally, multi-allelic, allele mismatched, and strand ambiguous SNPs alongside SNPs with duplicated positions, missing test statistics, and indels were removed[57].

## LD-Score Regression

SNP based heritability ($h^2_{SNP}$) and genetic correlations ($r_{g\,SNP}$) between all cleaned MD and CMD GWAS summary statistics were estimated using linkage-disequilibrium score regression (LDSC)[58,59] version 1.0.1 using authors' protocols.

## Cumulative Incidences

We estimated the cumulative incidence of all MDs and CMDs, which can be interpreted as the number of cases happening before a specific age. The cumulative incidences were estimated for the general population, individuals with one or more full siblings diagnosed with the same disorder, and individuals with one or more parents diagnosed with the cross-disorder (e.g., the cumulative incidence of ADHD for individuals with at least one parent diagnosed with type-2 diabetes). We expected the distribution of individuals into these three categories to be associated with birth year. Thus, to control for substantial changes over time in the underlying incidence, diagnoses (e.g., shifting of ICD systems), data availability, and registration (e.g., use of inpatient diagnoses up to 1995/2000 and in- and out-patient diagnoses subsequently), all cumulative incidences were estimated stratifying on the year of birth using the Nelson-Aalen estimator, which can utilise censored, competing risks, and incomplete data[19]. Next, we estimated the additive heritability ($h^2$) and genetic correlation ($r_g$) under the liability threshold model based on the cumulative incidence as a function of pedigree relatedness following procedures described by Wray and Gottesman[21,60,61]. In short, the liability threshold model assumes that disease liability underlying the disease status is normally distributed, $Z \sim N(0,1)$, and individuals with the disorder must therefore have surpassed a liability threshold[62,63]. Given the normal distribution theory, the liability threshold of a given disorder can be estimated from the population that are affected in their lifetime (lifetime risk). All analyses were done in R v4.2.1 using the cmprsk v2.2 package.

## Heritability ($h^2$)

Using the full available register data (no restriction of birth year), the heritability of liability of disorders was calculated by deriving the general population- (e.g., risk of ADHD in the population) and full-sibling familial risk (e.g., risk of ADHD when having a full-sibling with ADHD) cumulative incidences for individuals born in the same calendar year (e.g., 1965, 1966, till 2016). Here, we use the cumulative incidence (general population and full-sibling risk) at the last observed time point as estimates of the proportion of the population born in the same calendar year that is affected in their lifetime resulting in estimates of heritability (Eqs. 1 and 2) for individuals born in the same

calendar year ($h^2_{year\,of\,birth}$).

$$\text{Heritability}\,(h^2) = \frac{T - T_R\sqrt{\left(1 - (1 - T/i)\left(T^2 - T_R^2\right)\right)}}{a_R\left(i + (i - T)T_R^2\right)} \quad (1)$$

$$\text{s.e}\left(h^2\right) = \frac{1}{a_R}\sqrt{\left[\frac{K^2}{y^2}\left(\frac{1}{i} - a_R h^2(i - T)\right)^2 + \frac{K_R^2}{i^2 y_R^2}\right]} \quad (2)$$

Where T = Liability threshold of the disease in the general population, $T_R$ = liability threshold of the disease based on affected family members, i = mean liability of disease in the population calculated as i = y/K; where K is the lifetime probability of disease in the population and y the height of the normal curve at threshold T, $a_R$ = additive genetic relationship between relatives, $K_R$ = the lifetime probability of disease in individuals with affected family members. Note that all estimates are derived for individuals born in the same calendar year.

## Genetic correlation ($r_g$)

In contrast to the $h^2$ estimation, for the genetic correlation, we restricted the birth window to individuals born between 1981 and 2005, using medical records up to 2012. The $r_g$ between disorders was calculated by deriving: the general population risk for both disorders (e.g., ADHD and T2D) and parent-offspring cross disorder familial risk (e.g., risk of ADHD when having a parent with T2D) cumulative incidences for individuals born in the same calendar year (e.g., 1981,1982 till 2005). In line with the $h^2$ estimation, we used the cumulative incidence at the last observed time point for each birth year for all three cumulative incidence functions (general population risk and cross-disorder familial risk). Using the $h^2$ of both disorders previously obtained we derived estimates of genetic correlations (Eqs. 3 and 4) per year of birth ($r_{g,year\,of\,birth}$).

$$\text{Genetic correlation}\,(r_g) = \frac{\left(\frac{T_c - T_{R_c}\sqrt{1 - (1 - T_f/i_f)\left(T_f^2 - T_{R_c}^2\right)}}{a_R\left(i_f + (i_f - T_f)T_{R_c}^2\right)}\right)}{\sqrt{h_c^2 h_f^2}} \quad (3)$$

$$\text{s.e}\left(r_g\right) = \frac{\frac{1}{a_R}\sqrt{\left[\frac{K_f^2}{y_f^2}\left(\frac{i}{i_f} - a_R r_{cf} h_c h_f\left(i_f - T_f\right)\right)^2 + \frac{1}{i_f^2}\left(\frac{K_{R_c}^2}{y_{R_c}^2} + \frac{K_c^2}{y_c^2}\right)\right]}}{\sqrt{h_c^2 h_f^2}} \quad (4)$$

Where $T_c$ and $T_f$ = liability threshold of disease c and f in the general population, $T_{R_c}$ = liability threshold of disease c in individuals with relatives with disease f, $i_f$ = mean liability of disease f in the population, $a_R$ = additive genetic relationship between relatives, $h_c^2$ and $h_f^2$ = heritability of diseases c and f, $K_f$ is the lifetime probability of disease f in the general population. Note that all estimates are derived for individuals' born in the same calendar year.

## Random effect inverse variance weighted model ($IVW_{random}$)

We obtain overall $h^2$ and $r_g$ estimates by weighing the individual $h^2_{year\,of\,birth}$ and $r_{g,year\,of\,birth}$ by the inverse of their sampling variance (Eqs. 5 and 6) using a random-effects model.

$$IVW_{random} = \frac{\sum_{k=1}^K \hat{\theta}_k w_k^*}{\sum_{k=1}^K w_k^*}; w_k^* = \frac{1}{s_k^2 + r^2} \quad (5)$$

$$s.e.\,(\text{IVW}_{\text{random}}) = \sqrt{\frac{1}{\sum_{k=1}^{K} w_k^*}} \qquad (6)$$

Where K = numbers of estimates, $s_k^2$ = variance of estimate k, $r^2$ = the variance of the distribution of true effect sizes, $\hat{\theta}_k$ = point estimate k, and $w_k^*$ = random-effects weight.

## Quantification of genetic and non-genetic factors

Under a bivariate liability threshold model, the phenotypic correlation ($r_P$) between two traits can be broken down to its (additive) genetic and non-genetic factors. This allows us to quantify and understand the contribution of the estimated genetic correlation and heritability to the level of comorbidity between MDs and CMDs, i.e., hazard ratios) reported by Momen et al. [1] which uses the same Danish register data.

$$\text{Relative risk}\,(\text{RR}) = \frac{1 - e^{(\text{HR} \times \log(1-r))}}{r} \qquad (7)$$

Where HR = hazard ratio reported by Moment et al. 2020, and r = rate of the disorder in the reference group derived by weighting the individual estimates (1981-2005 using medical records up to 2012) by the inverse of their sampling variances.

$$\text{Odds ratio}\,(\text{OR}) = \frac{(1-p) \times \text{RR}}{1 - (\text{RR} \times p)} \qquad (8)$$

Where p = incidence of the disorder in the nonexposed group (here p = r) and RR = the calculated relative risk.

$$\text{Phenotypic correlation}\,(r_p) = \frac{\text{OR}^{\frac{\pi}{4}} - 1}{\text{OR}^{\frac{\pi}{4}} + 1} \qquad (9)$$

Where OR = odds ratio estimated as a function of relative risk.

## Decomposition of comorbidity (phenotypic correlation)

The phenotypic correlation can be expressed as the function of the genetic and non-genetic component

$$\text{Phenotypic correlation}\,\left(r_p\right) = r_g \sqrt{h_c^2 h_f^2} + r_e \sqrt{\left(1 - h_c^2\right)\left(1 - h_f^2\right)};$$

$$\text{Genetic component}\,(G) = r_g \times \sqrt{h_c^2 \times h_f^2} \qquad (10)$$

$$\text{Non-genetic component}\,(E) = r_p - G \qquad (11)$$

Where $r_p$ = tetrachoric correlation derived from the HR, $r_g$ = the genetic correlation estimates and $r_e$ the environmental correlation estimates between disorder c and f, $h^2$ the heritability estimates for c and f. 95% CIs for G and E were derived using both the upper and lower 95% CIs of $r_g$ and $r_p$. Note to estimate $G_{\text{SNP}}$ and $E_{\text{SNP}}$ replace $r_g$ and $h^2$ estimates by SNP based estimates $r_{g\,\text{SNP}}$ and $h^2_{\text{SNP}}$.

## Reporting summary

Further information on research design is available in the Nature Portfolio Reporting Summary linked to this article.

## Data availability

The data generated in this study are provided in the Supplementary data file. GWAS summary statistics including stroke (subtypes), CAD, aneurysms and HF were obtained through multiple public repositories (supplementary data 6). GWAS summary statistics containing participants of the VA Million Veterans Programme (e.g., T2D, venous thromboembolism, and peripheral artery disease) were provided after approval was granted by the National Institute of Health (*project #26508*). GWAS summary statistics for ADHD, AN, ASD, BD, and MDD excluding iPSYCH participants (except SCZ which does not contain iPSYCH samples) were kindly provided through their respective PGC consortium. iPSYCH only GWAS summary statistics for MDs were downloaded from internal iPSYCH servers and are available on request.

## Code availability

Danish register data was stored on a PostgreSQL 13.3 database server information was extracted using the psql 16.2 database client. All register-based analyses were done in R v4.2.1 using the cmprsk v2.2 package. All analyses performed on genotype data (GWAS summary statistics) were performed using linkage-disequilibrium score regression (LDSC) version 1.0.1. GWAS summary statistics were cleaned using the cleansumstats pipeline[57]. Mathematical formulas to calculate $h^2$ and $r_g$ were taken from[21].

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

## Acknowledgements

This research was supported by: the European Union's Horizon 2020
Research and Innovation Programme: the "predicting comorbid cardi-
ovascular disease in individuals with mental disorder by decoding dis-
ease mechanisms" project (CoMorMent, grant number 847776, to J.M.,
J.B., A.B., U.V., D.M., Y.L., T.W., O.A., and F.F.); the Danish National
Research Foundation (grant number DNRF148); the US National Insti-
tutes of Health study on extreme MDD (R01 MH123724, to J.M., J.R.S.,
Y.L., and A.B.); the European Research Council (grant agreement ID
101042183, to Y.L.),.the Sir Henry Wellcome Postdoctoral Fellowship
(Reference 213674/Z/18/Z, to D.H.); the Research Council of Norway
(RCN grants 324499, 324252, 223273, to O.A.); the Stiftelsen Kristian
Gerhard Jebsen (grants SKGJ-MED-008 and SKGJ-MED-021, to O.A).; the
Laureate Grant Award from the Novo Nordisk Foundation (Grant No:
NNF22OC0071010, to N.M.); European Research Council Consolidator
grant (StressGene, Grant nr. 726413 to U.V.), and Icelandic Research
fund (to U.V.). The iPSYCH Initiative is funded by the Lundbeck Foun-
dation (Grant Nos. R268-2016-3925, R102-A9118, and R155-2014-1724),
the Mental Health Services Capital Region of Denmark, University of
Copenhagen, Aarhus University, and the University Hospital in Aarhus.
Genotyping of iPSYCH samples was supported by grants from the
Lundbeck Foundation, the Stanley Foundation, the Simons Foundation
(Grant No. SFARI 311789), and the National Institutes of Mental Health
(Grant No. 5U01MH094432-02). The iPSYCH Initiative uses the Danish
National Biobank resource that is supported by the Novo Nordisk
Foundation. We like to thank the Psychiatric Genomics Consortium
working groups (major depressive disorder, attention-deficit/hyper-
activity disorder, autism spectrum disorder, schizophrenia, bipolar dis-
order, and eating disorders) for contributing genome-wide association
summary statistics data. We like to thank all the senior members of the
iPSYCH consortium: Preben Bo Mortensen, Anders Børglum, David
Hougaard, Thomas Werge, and Merete Nordentoft. Finally, we would like
the thank Hakon Heimer for editing the final version of the manuscript
and Mischa Lundberg for all-round moral support.

## Author contributions

JMeijsen, T.W., and A.B. conceived the study. JMeijsen, R.S., A.B., D.M.,
O.P., M.D.K., and G.A .contributed to the study design. JMeijsen, K.H.,
J.G., A.I., and R.Z. performed the literature search, programming and/or
data analyses. JMeijsen, A.B., S.W., M.D.K., J.S., O.A., J.B., O.P., JMcGrath,
U.V., N.M., and T.W. contributed to data interpretation. F.F., W.Y., D.H.,
Y.L., A.B., and T.W. provided access to data. JMeijsen, A.B., T.W., and
S.W. wrote the initial draught. O.A.A. and TW obtained the funding.

## Competing interests

NM receives an honorarium to serve as associate editor on the European
Eating Disorders Review board. O.A.A. is a consultant to Corteechs.ai
and Precision Health AS, and received speaker's honorarium from
Janssen, Sunovion, Otsuka and Lundbeck. U.A.V. declares receiving
support from EPA2023, ISTSS2022 as keynote speaker, and serves on a
NordForsk expert committee on Long COVID. All other authors declare
no competing interests.

## Additional information

**Supplementary information** The online version contains
supplementary material available at

Joeri Meijsen or Alfonso Buil.

**Peer review information** *Nature Communications* thanks the anon-
ymous reviewers for their contribution to the peer review of this work. A
peer review file is available.

¹Institute of Biological Psychiatry, Mental Health Center Sct. Hans, Mental Health Services Copenhagen University Hospital, Roskilde, Denmark. ²Unit of
Integrative Epidemiology, Institute of Environmental Medicine, Karolinska Institutet, Stockholm, Sweden. ³Department of Evolutionary Biology, Ecology and
Environmental Sciences, University of Barcelona, Barcelona, Spain. ⁴Center for Eating and feeding Disorders research, Psychiatric Centre Ballerup, Mental
Health Services in the Capital Region of Denmark, Copenhagen, Denmark. ⁵Department of Science and Environment, Roskilde University, Roskilde, Denmark.
⁶Social, Genetic and Developmental Psychiatry Centre, Institute of Psychiatry, Psychology & Neuroscience, King's College London, London, UK. ⁷Department
of Medical Epidemiology and Biostatistics, Karolinska Institutet, Stockholm, Sweden. ⁸Centre of Public Health Sciences, Faculty of Medicine, University of
Iceland, Reykjavík, Iceland. ⁹Department of Epidemiology, Harvard T.H. Chan School of Public Health, Boston, MA, USA. ¹⁰Department of Clinical Epide-
miology, Aarhus University and Aarhus University Hospital, Aarhus, Denmark. ¹¹Queensland Centre for Mental Health Research, Brisbane, Australia.
¹²Queensland Brain Institute, The University of Queensland, Brisbane, Australia. ¹³National Centre for Register-based Research, Aarhus University,
Aarhus, Denmark. ¹⁴Great Ormond Street Institute of Child Health, University College London, London, UK. ¹⁵NORMENT Centre, Division of Mental Health and
Addiction, Oslo University Hospital, Oslo, Norway. ¹⁶Institute of Clinical Medicine, University of Oslo, Oslo, Norway. ¹⁷KG Jebsen Centre for Neurodevelop-
mental disorders, Institute of Clinical Medicine, University of Oslo, Oslo, Norway. ¹⁸Department of Clinical Medicine, University of Copenhagen,
Copenhagen, Denmark. ✉e-mail: joeri.jeroen.meijsen@regionh.dk; alfonso.buil.demur@regionh.dk

