## [Peer Review File · Nature Communications]

Quantifying the relative importance of genetics and environment on the comorbidity between mental and cardiometabolic disorders using 17 million ScandinaviansREVIEWER COMMENTS

Reviewer #1 (Remarks to the Author):

This paper is investigating the communality of a number (6) of mental disorders (MDs) and several (14) cardiometabolic diseases (CMDs), to estimate their heritabilities and, more importantly, the correlation in occurrence of MD and CMD pairs and the degree to which these correlations can be attributed to either genetic or environmental components. The authors use the fairly complete population-based cohorts of Denmark and Sweden, which have kept track of major diseases and hospitalizations of individuals living in these countries over decades. The authors conclude that these diseases (both MDs and CMDs) are heritable, that pairs of these diseases tend to be positively correlated for the most part (with some exceptions), that correlations overall appear to be driven more by shared environmental contributions than by shared genetic factors (again with some exceptions), but that each disease pair investigated (84 in all, being equal to 6 MDs times 14 CMDs) is unique in its characteristics and should be viewed by itself.

The available population-based databases are major assets to this paper, the analyses of which would not otherwise have been possible. The investigators have performed a substantial number of analyses tailored to their unique datasets. I do not question the validity of the analytical approaches and findings. The results overall are not really surprising in direction of effect or in magnitude of effect, but it is of value that this paper addresses quite a number of pairwise comparisons between MDs and CMDs comprehensively and adds to the literature for an overall comparison between these two disease classes and the specific pairs investigated. Given that the paper used unique and highly valuable datasets, these estimates, while perhaps not unexpected, are important and not merely redundant, replicatory datapoints.

The biggest drawback of the manuscript is perhaps that the findings are quite general, largely as expected, and one is not quite sure exactly what to do with all of the estimates of pairwise phenotypic, genetic, and environmental correlations. They could be used as impetus for examining in more detail specific disease pairs, but many of these studies are already being conducted based on prior data suggesting an interesting and important link between two paired conditions.

The authors should explain exactly why these conditions were selected, and why some diseases were collapsed into a single category or kept apart. Where these simply the most common conditions found in the Danish database? Among the MDs, why were bipolar disorder (BP) and major depressive disorder (MDD) combined as affective disorders (AFF)? Among the CMDs, what about obesity or dyslipidaemia, which are important components of cardiometabolic diseases (CMDs)? Is their absence related to the fact that they may not have led to any hospitalizations?

I am unsure exactly what the comparison among register-based results and those based on SNPs adds to the paper. This may be methodologically interesting but does not appear to be central to the paper's main purpose. If the authors want to keep this section, then they should discuss why the SNP-based results tend to be quite a bit weaker overall. The reasons are somewhat obvious but deserve to be explained in the paper anyway, as not all readers will be knowledgeable in this area.

Re. Supplementary Figure 2 (erroneously labeled Figure 2 in the supplementary figures file) and the related text (lines 155 and onwards) in the manuscript: This topic is somewhat obvious and has been covered well in many publications. No simulation study is necessary. The equation $\rho_p = \sqrt{h_1^2} \sqrt{h_2^2} \rho_g + \sqrt{(1-h_1^2)} \sqrt{(1-h_2^2)} \rho_e$ explains these issues well in my view. A given trait's heritability also puts a limit to the potential utility of biomarkers as surrogates for that trait. See the "endophenotype ranking values" by Glahn and Blangero. I feel that some passages of the manuscript should be re-written or streamlined with this equation and relevant references in mind.

Minor comments:

"When the comorbidity is driven mainly by environmental factors, clinical approaches will be limited to interventions in the environment.": This seems to be an overstatement. E.g., let us assume the risk of a mental and a CMD are both influenced by, say, salt intake. Sure, intervening in the environment by reducing salt intake would likely be helpful. But so may be a drug that leads to a lowered salt uptake, an increased salt excretion, and/or an increased salt tolerance. I get the motivation behind the sentence, but the statement is perhaps too strong and/or simplistic.

"These observations are consistent with ADHD being highly heritable and with significant genetic-correlation between ADHD and type-2 diabetes.": While I do not disagree with this statement, Figure 1A is also consistent with lots of different scenarios, such as no heritability and significant environmental correlation. In fact, the fact that the cumulative incidence functions vary substantially across birth years suggest an important non-genetic component (though there are other contributors to which the authors allude). I suggest leaving this statement out here, as it seems to prematurely predict what later observations, based on different types of analyses, show.

"We applied this analytical approach ..." (line 145): What analytical approach?

"Bonferroni $p < 5.98 \times 10^{-4}$ " on line 150 and "Bonferroni $p < 5.95 \times 10^{-4}$ " on lines 192 as well as 194: These should be consistent, and the first instance likely is a typo, as $0.05/84 = 5.95 \times 10^{-4}$.

Figure 1 (and similar figures): The 95% confidence intervals are essentially not visible and make the plots less clear. I suggest leaving them out, or perhaps include them only with the oldest and the youngest birth year, but not for each birth year. For what it is worth, I enjoyed seeing these curves (especially panels A and B). I wonder whether they would be useful as supplementary material (either as figures and/or tables) for all examined traits.

Re. Figures 2 - 4: In Figure 2, I question whether the ordering of traits (among MDs and CMDs, respectively) by heritability is helpful. Perhaps some order based on the nature of these traits, and how similar/related they are to one another, would facilitate understanding. E.g., list next to on both forms of ICA (unruptured and all). After all, the estimated heritabilities per se are not an organizing pillar of this paper. And whatever order is chosen in Figure 2 should be maintained in Figure 3 and 4 for consistency. Also, why are only 5 out of 6 MDs included in Figure 4 and only 7 out of 14 CMDs?

Reviewer #2 (Remarks to the Author):

Thank you for the opportunity to review the manuscript. Dr Meijssen et al investigated the genetic and environmental contributions underlying the observed comorbidity between six MDs and 14 CMDs and reported that genetic factors contributed about 50% to the comorbidity of schizophrenia, affective disorders, and autism spectrum disorder with CMDs, whereas the comorbidity of attention-deficit/hyperactivity disorder and anorexia with CMDs was mainly or fully driven by environmental factors. This work shed light on preventing and treating CMD in individuals with mental disorders. The statistical methods were corrected and the manuscript was generally well-written. I only have one concern. The authors reported that the age at first diagnosis for CVD was very young, for example, 1 year old? Please recheck the data and perform more sensitivity analyses if necessary.

Reviewer #3 (Remarks to the Author):

The authors should be commended for this extensive body of work that seeks to advance understanding of mental disorders and their link with cardiometabolic diseases.

Major comments

1. Ancestry is a major factor to consider when conducting any genetic analyses. There is no mention of the ancestry of the population data. One could assume that all individuals are European ancestry (as nothing to the contrary is stated), however that is problematic, particularly given recent history of population movements (I am more aware of these for Sweden than Denmark admittedly). Has the genetic data been used to confirm the ancestry of the populations? More information about this is required (including justification if this has not been attempted).

2. What is the rationale for considering depression and bipolar combined (as affective disorders) and then bipolar separately but not depression separately? I can see the logic for the combined affective disorder, given that the diagnoses are overlapping and that there can be some misdiagnoses (with unipolar depression being one end of the spectrum and bipolar the other end, with many clinical cases with presentation falling in the middle). But would it not be more logical to consider unipolar depression separately (as a more distinct phenotype) as well as the combined affective disorder? Or bipolar, depression and affective disorder? This is particularly interesting given the recent awareness of increasing prevalence of depression worldwide.

3. A similar point can be made considering obesity, as a conspicuously missing cardiometabolic phenotype. What is the rationale for not including this, as it is perhaps the most obvious (from observational studies as well as biological plausibility) link between mental disorders and the subsequent CMD investigated? If there is good rationale, this should be made clear.

4. Regarding limitations, to what extent does the use of electronic records from hospital treatment influence the results (compared to say general practice diagnoses)?

5. The limitations suggest that the results might not translate well to "non-Scandinavian" countries, hinting that within Scandinavia, perhaps the environmental/genetic factors might be similar across other Scandinavian countries. Is this a fair assumption?

Minor comments

1) With the methods being presented after the results, please check that abbreviations are defined at first use (SNP, CMD)

2) Consistency should be improved: population-complete genealogies or near-complete? Cardiovascular and metabolic or cardiometabolic?

3) Please define abbreviations in supplementary table footnotes

4) Please check for typos: line 156, neglectable vs negligible? Line 263, recoding or recording?

REVIEWER COMMENTS

Reviewer #1 (Remarks to the Author):

This paper is investigating the communality of a number (6) of mental disorders (MDs) and several (14) cardiometabolic diseases (CMDs), to estimate their heritabilities and, more importantly, the correlation in occurrence of MD and CMD pairs and the degree to which these correlations can be attributed to either genetic or environmental components. The authors use the fairly complete population-based cohorts of Denmark and Sweden, which have kept track of major diseases and hospitalizations of individuals living in these countries over decades. The authors conclude that these diseases (both MDs and CMDs) are heritable, that pairs of these diseases tend to be positively correlated for the most part (with some exceptions), that correlations overall appear to be driven more by shared environmental contributions than by shared genetic factors (again with some exceptions), but that each disease pair investigated (84 in all, being equal to 6 MDs times 14 CMDs) is unique in its characteristics and should be viewed by itself.

The available population-based databases are major assets to this paper, the analyses of which would not otherwise have been possible. The investigators have performed a substantial number of analyses tailored to their unique datasets. I do not question the validity of the analytical approaches and findings. The results overall are not really surprising in direction of effect or in magnitude of effect, but it is of value that this paper addresses quite a number of pairwise comparisons between MDs and CMDs comprehensively and adds to the literature for an overall comparison between these two disease classes and the specific pairs investigated. Given that the paper used unique and highly valuable datasets, these estimates, while perhaps not unexpected, are important and not merely redundant, replicatory datapoints.

We appreciate that the reviewer recognises our effort to add to the literature and not perform “merely redundant, replicatory datapoints”, but to go beyond that.

The biggest drawback of the manuscript is perhaps that the findings are quite general, largely as expected, and one is not quite sure exactly what to do with all of the estimates of pairwise phenotypic, genetic, and environmental correlations. They could be used as impetus for examining in more detail specific disease pairs, but many of these studies are already being conducted based on prior data suggesting an interesting and important link between two paired conditions.

The motivation for this work is the wide interest in understanding the causes underlying the comorbidity between mental and cardiometabolic disorders. Recent literature offers many publications on the genetic overlap between MD and CVD, however most of those publications obtain their estimates from summary statistics from genome-wide association studies (GWAS) and suffer from severe limitations.

We updated the introduction of the manuscript to better highlight these points.

“Previous studies have found a substantial genetic overlap between CMD and MD^{12–15}, which suggests that genetics contribute to the MD-CMD comorbidity. However, these studies obtain their estimates for genetic overlap from summary statistics of genome-wide association studies (GWAS), a strategy that suffers from serious limitations. First, these studies are affected by extreme sampling of cases suffering from comorbidities other than the index diagnosis under study, and of control participants recruited among overly healthy subjects, and thus, they are genuinely unrepresentative of the background population¹⁶. Second, these GWAS focus on specific disorders, and different populations and methods, but a systematic analysis for a large spectrum of disorders in the same population is lacking. Third, the estimates (LDSC, GWAS, PGS) are sensitive to the underlying assumptions of the statistical model and the linkage disequilibrium panel used to calculate them¹⁷. Moreover, these previous studies report genetic correlations as a measure of shared genetic effects between two disorders but do not investigate the relative importance of genetic and environmental factors on the comorbidity between disorders.”

In this work, we overcame some of those limitations by conducting a unique study in terms of both the data available and the analytical approach. We would like to highlight four aspects that make our study a step forward in the state of the art:

- *The use of national registers with millions of individuals followed for several decades provides a comprehensive view of the problem.*
- *The study in one nation and replication in another provides a high degree of confidence in our estimates.*
- *The estimate of the relative contribution of genetic and environmental factors to the observed comorbidity between MD and CMD goes beyond the usual estimates of genetic correlations.*
- *The comparison of register-based vs SNP-based estimates in the same population addresses a fundamental methodological question and highlights the advantages of performing genetic analyses in national health register samples.*

We would like to reiterate what we mention in the Discussion section of the manuscript:

“A strength of this study is the use of nationwide registers in Denmark and Sweden. These countries share many similarities in history, (genetic) ancestry, cultural characteristics, and their government funded healthcare system. However, clinical variation such as different recording of diagnosis periods, use of different ICD versions²⁸, and varying diagnostic practices have resulted in country-specific trends over time. Despite this, replication analysis of heritability and genetic correlation estimates derived using the larger Swedish registers showed no significant difference from the Danish estimates. The high transferability between the two genealogies adds validity to the findings on heritability and genetic correlations. To complement our register analysis, we computed SNP-based estimates of heritability and genetic correlations leveraging molecular genetic data on the Danish population-based sample (iPSYCH 2012 cohort). To our knowledge this

is the first comparison of heritability and genetic correlations based on genotypes and register data carried out in the same population."

The authors should explain exactly why these conditions were selected, and why some diseases were collapsed into a single category or kept apart. Where these simply the most common conditions found in the Danish database? Among the MDs, why were bipolar disorder (BP) and major depressive disorder (MDD) combined as affective disorders (AFF)? Among the CMDs, what about obesity or dyslipidaemia, which are important components of cardiometabolic diseases (CMDs)? Is their absence related to the fact that they may not have led to any hospitalizations?

We thank the reviewer for their questions. For your convenience we separated the main text and answered the questions individually below.

Question 1: The authors should explain exactly why these conditions were selected, and why some diseases were collapsed into a single category or kept apart. Where these simply the most common conditions found in the Danish database?

One of our aims was to utilise the unique opportunity to compare genetic- and register-based genetic estimates of almost the identical set of individuals within an entire country. Therefore, we set out to keep as many variables as possible similar such as: birth cohort, observation time, and phenotypes. To keep the phenotypes similar, we used the six mental disorders defined and used for GWAS analysis of iPSYCH 2012 data by AJ Schork et al, 2019. These disorders are also the most well documented, well known, and common mental disorders occurring in the population. The cardiometabolic disorders were selected based on a.) availability of GWAS summary statistics on publicly accessible databases, and b) all disorders came from a wide range of prevalence in the population (low, medium, and high prevalence).

We have added this to the case definition section:

“Mental disorders were previously defined and used for GWAS analysis of iPSYCH 2012 data by Schork et al, 2019. These disorders represent the most well documented, well known and most common mental disorders occurring in the population. The cardiometabolic disorders were selected based on a.) common in the population i.e., high prevalence or b.) less common prevalence i.e., low prevalence and c.) selected disorder had available GWAS summary statistics in any public available repository.”

Question 2: Among the MDs, why were bipolar disorder (BP) and major depressive disorder (MDD) combined as affective disorders (AFF)?

We acknowledge that it is interesting to separate AFF into BP and MDD. We have calculated the genetic correlations between MDD and the CMDs and report them here (Table below). We observed no significant difference with the AFF results mentioned throughout the paper. These results support our assumption that MDD backs up the bulk of AFF diagnoses and therefore the results are highly similar.

	Denmark								Sweden			
	AFF				MDD							
	rg	se	l95	u95	rg	se	l95	u95	rg	se	l95	u95
AP	0.25	0.04	0.17	0.32	0.24	0.01	0.16	0.32	0.15	0.02	0.06	0.24
AIS	0.30	0.06	0.18	0.41	0.30	0.02	0.19	0.41	0.24	0.02	0.12	0.36
AS	0.30	0.06	0.18	0.43	0.31	0.02	0.19	0.44	0.27	0.02	0.14	0.40
AF	0.08	0.05	-0.01	0.17	0.08	0.01	0.00	0.17	0.08	0.02	-0.01	0.17
CIHD	0.19	0.04	0.12	0.26	0.18	0.01	0.10	0.26	0.12	0.02	0.04	0.21
CAD	0.23	0.04	0.15	0.30	0.22	0.01	0.15	0.30	0.17	0.02	0.04	0.29
HF	0.24	0.06	0.12	0.36	0.25	0.02	0.13	0.37	0.21	0.02	0.11	0.31
HYP	0.16	0.03	0.11	0.21	0.16	0.01	0.10	0.21	0.16	0.01	0.10	0.21
ICA (R & U)	0.18	0.07	0.05	0.31	0.18	0.03	0.03	0.34	0.23	0.02	0.09	0.37
ICA (U)	0.14	0.11	-0.08	0.36	0.23	0.08	-0.05	0.53	0.08	0.03	-0.06	0.23
PAD	0.27	0.08	0.11	0.43	0.27	0.02	0.11	0.42	0.22	0.01	0.13	0.32
SAH	0.20	0.08	0.03	0.36	0.15	0.02	0.00	0.31	0.32	0.02	0.12	0.52
T2D	0.20	0.04	0.13	0.27	0.20	0.01	0.13	0.28	0.22	0.01	0.16	0.27
VET	0.24	0.05	0.17	0.32	0.24	0.02	0.14	0.34	0.15	0.01	0.06	0.23
Obesity	0.48	0.02	0.36	0.60	0.48	0.02	0.35	0.60	0.15	0.02	0.06	0.24

We added this statement (below) to the discussion and added the results in supplementary table S4B. We calculated the *rg* between MDD and all CMDs and observed no significant difference with the AFF results supporting our assumption that MDD makes up the majority of AFF diagnoses (Table S4B).”

Question 3: Among the CMDs, what about obesity or dyslipidaemia, which are important components of cardiometabolic diseases (CMDs)? Is their absence related to the fact that they may not have led to any hospitalizations?

We agree that obesity is an interesting phenotype for this type of research, but we have been reluctant to use it because we suspect it is not well recorded in the registers. On the one hand, as a general norm, only extreme obesity cases are recorded in the registers, and on the other hand, it can be that many non-extreme cases are recorded, indicating that the patient is in the hospital for another reason. For example, a patient with SCZ who becomes obese because of secondary effects of medication can get the obesity diagnosis recorded while visiting the hospital related to the SCZ. That implies that the recorded obesity diagnosis can be biased toward people with other conditions, including mental conditions, which will bias the genetic correlation estimates. However, given that two out of three reviewers asked about it, we decided to present the results with the disclaimer that we should be aware of the potential biases. We have identified over 350K diagnosed individuals with obesity in the registers and included register-based genetic estimates in the paper (Figures 2 and 3 and supplementary table S2 and S4) and added the results here as well for your convenience.

Genetic correlation between obesity and mental disorders												
	Denmark				Sweden				Meta-analysis			
	rg	se	I95	u95	rg	se	I95	u95	rg	se	I95	u95
ADHD	0.43	0.02	0.35	0.50	0.51	0.02	0.41	0.62	0.47	0.04	0.38	0.55
ASD	0.11	0.02	0.01	0.21	0.31	0.02	0.23	0.40	0.52	0.04	0.43	0.60
ANO	-0.23	0.03	-0.39	-0.07	-0.25	0.03	-0.41	-0.10	-0.24	0.02	-0.29	-0.20
AFF	0.48	0.02	0.36	0.60	0.56	0.03	0.41	0.71	0.21	0.10	0.01	0.41
BIP	0.32	0.04	0.13	0.50	0.30	0.03	0.18	0.42	0.31	0.02	0.26	0.35
SCZ	0.27	0.03	0.14	0.39	0.08	0.06	-0.17	0.32	0.18	0.10	-0.02	0.38

We excluded dyslipidaemia from this work since, at the time of gathering GWAS data, there were no large GWAS studies available to compare our register-based results to. Furthermore, we found that the ICD8 codes for dyslipidaemia were not adequate, and we preferred to have both ICD8 and 10 codes for each CMD.

I am unsure exactly what the comparison among register-based results and those based on SNPs adds to the paper. This may be methodologically interesting but does not appear to be central to the paper's main purpose. If the authors want to keep this section, then they should discuss why the SNP-based results tend to be quite a bit weaker overall. The reasons are somewhat obvious but deserve to be explained in the paper anyway, as not all readers will be knowledgeable in this area.

We added this paragraph to the discussion to make our case on the relevance and timeliness of this analysis. "The explosion of published GWAS was followed by an explosion of papers estimating the heritability and genetic correlations between diseases based on GWAS summary stats. Even though it is known and accepted that the SNP-based estimates are largely underestimates of the real values, SNP-based estimates are becoming the standard. Here we took advantage of the unique opportunity to compare SNP-based and register-based estimates of heritability and genetic correlations using almost the same set of individuals within an entire country. To our knowledge this is the first time that such a comparison has been done in a large sample. Our results confirm the fact that SNP-based heritability estimates capture only a small fraction of the total heritability and reveal that this fact has important consequences when decomposing genetic and environmental factors between two diseases."

Re. Supplementary Figure 2 (erroneously labeled Figure 2 in the supplementary figures file) and the related text (lines 155 and onwards) in the manuscript: This topic is somewhat obvious and has been covered well in many publications. No simulation study is necessary. The equation $\rho_p = \sqrt{h_1^2} \sqrt{h_2^2} \rho_g + \sqrt{(1-h_1^2)} \sqrt{(1-h_2^2)} \rho_e$ explains these issues well in my view. A given trait's heritability also puts a limit to the potential utility of biomarkers as surrogates for that trait. See the "endophenotype ranking values" by Glahn and Blangero. I feel that some passages of the manuscript should be re-written or streamlined with this equation and relevant references in mind.

We agree with the reviewer, it is obvious in the light of the equation. However, our experience has taught us that this equation and its consequences are unknown to many people working in genetics and most clinicians. Most papers presenting genetic correlations do not interpret those estimates together with the heritability, and we believe that the idea that a large genetic correlation between two traits means a large effect of genetics in its comorbidity is widely spread among clinicians, and a substantial number of geneticists. That is why, acknowledging that it is not a new concept, we still think that our emphasis on explaining the concept can be very useful to the readers of Nature Communications. Nevertheless, we take the reviewer's advice and add some context and references to the manuscript. We also thank the reviewer for noticing the mislabelling of the mentioned figure and we have now corrected that.

We wrote this in Register heritability (h^2) and genetic correlation (r_g) estimates in the results section:

“It is well known that the comorbidity between two diseases depends not only on their genetic and environmental correlations but also on their heritability^{22,23}. Genetic correlation alone, while estimating the degree of overlap of the genetic architectures of two disorders, may account for only a negligible fraction of the observed comorbidity. This scenario may occur for disorders with high genetic correlations when the heritability of one or both disorders is modest, as shown in simulated data in Figure S2. To determine the relative contributions of heritable genetic variants and environmental risk factors to MD-CVD comorbidities, we applied quantitative genetic methods that combine the estimates of heritability and genetic correlation for each pair of disorders.”

Minor comments:

“When the comorbidity is driven mainly by environmental factors, clinical approaches will be limited to interventions in the environment.”: This seems to be an overstatement. E.g., let us assume the risk of a mental and a CMD are both influenced by, say, salt intake. Sure, intervening in the environment by reducing salt intake would likely be helpful. But so may be a drug that leads to a lowered salt uptake, an increased salt excretion, and/or an increased salt tolerance. I get the motivation behind the sentence, but the statement is perhaps too strong and/or simplistic.

The reviewer made a valid point and we toned down our statement in the manuscript.

“When the comorbidity is driven mainly by environmental factors, clinical approaches will prioritize interventions in the environment. However, when genetics plays an important role, therapeutic approaches could also benefit from precision medicine tools such as genetic risk prediction models.”

“These observations are consistent with ADHD being highly heritable and with significant genetic-correlation between ADHD and type-2 diabetes.”: While I do not disagree with this statement, Figure 1A is also consistent with lots of different scenarios, such as no heritability and significant environmental correlation. In fact, the fact that the cumulative incidence functions vary substantially across birth years suggest an important non-genetic component (though there are other contributors to which the authors allude). I suggest leaving this statement out here, as it seems to prematurely predict what later observations, based on different types of analyses, show.

We thank the reviewer for this comment. It is true that the variation in CIF across years (Figure 1A) is compatible with many scenarios and, by itself, does not say anything about genetic or environmental effects. It is when we look at Figure 1A and 1B and see the difference in incidence between the general population and the incidence in individual with an affected relative, that the importance of genetic effects becomes evident. In figures 1A,B,C we tried to explain two different concepts: 1) The larger incidence of a disease in individuals with affected relatives than in the general population implies a genetic effect; and 2) Changes in medical practice and the health recording system over time result in changes in CIF. Reviewers comment convinced us that our current presentation is confusing, and we decided to rewrite the paragraph to clarify our point.

New paragraph in the paper:

“Notably, the CIF for individuals with a sibling with ADHD (Figure 1B), as well as the CIF for individuals with a parent diagnosed with Type-2 diabetes (Figure 1C) are markedly increased relative to the CIF of ADHD for the entire population (Figure 1A). When comparing year of birth matched cumulative incidences reported in Figure 1A and 1B, we see a substantial difference in ADHD incidence in the general population (Figure 1A) versus individuals with an affected relative (Figure 1B). This difference highlights the importance of genetic effects while trying to account for substantial non-genetic components. These observations are consistent with ADHD being highly heritable and with significant genetic-correlation between ADHD and type-2 diabetes²⁰. All Danish CIFs are listed in Table S1A-G.”

“We applied this analytical approach ...” (line 145): What analytical approach?

We have now added some detail by rewriting the sentence to this:

“We applied the analytical approach reported by Wray and Gottesman”

“Bonferroni $p < 5.98 \times 10^{-4}$ ” on line 150 and “Bonferroni $p < 5.95 \times 10^{-4}$ ” on lines 192 as well as 194: These should be consistent, and the first instance likely is a typo, as $0.05/84 = 5.95 \times 10^{-4}$.

Thank you for noticing this and we have now corrected it; indeed, it was a typo. Because we added an extra phenotype to the paper, we changed all mentions of Bonferroni to $0.05/90 = 5.55 \times 10^{-4}$

Figure 1 (and similar figures): The 95% confidence intervals are essentially not visible and make the plots less clear. I suggest leaving them out, or perhaps include them only with the oldest and the youngest birth year, but not for each birth year. For what it is worth, I enjoyed seeing these curves (especially panels A and B). I wonder whether they would be useful as supplementary material (either as figures and/or tables) for all examined traits.

We thank the reviewer for their questions. For your convenience we separated the main text and answered the questions individually below.

Question 1: Figure 1 (and similar figures): The 95% confidence intervals are essentially not visible and make the plots less clear. I suggest leaving them out, or perhaps include them only with the oldest and the youngest birth year, but not for each birth year.

We tried to include the reviewer’s helpful suggestion. However, including the confidence intervals makes the figures generally less clear. We believe this outweighs the fact that excluding the confidence intervals would, in our opinion, remove vital information about sample size and we couldn’t in good conscience report purely point estimates.

Question 2: For what it is worth, I enjoyed seeing these curves (especially panels A and B). I wonder whether they would be useful as supplementary material (either as figures and/or tables) for all examined traits.

We are happy to hear that this figure was well received, and we would like to thank the reviewer for these kind words. We acknowledge that these figures could be helpful for clinicians and other researchers, and we considered adding all cumulative incidence plots as supplementary material, but we opted to provide the raw cumulative incidences of Denmark instead as supplementary tables 1A-1G. This would allow any clinician or researcher to directly search for risk estimates and derive their own figures if needed. In line with Danish law we set all count ≤ 5 to $N \leq 5$.

Re. Figures 2 - 4: In Figure 2, I question whether the ordering of traits (among MDs and CMDs, respectively) by heritability is helpful. Perhaps some order based on the nature of these traits, and how similar/related they are to one another, would facilitate understanding. E.g., list next to on both forms of ICA (unruptured and all). After all, the estimated heritabilities per se are not an organizing pillar of this paper. And whatever order is chosen in Figure 2 should be maintained in Figure 3 and 4 for consistency. Also, why are only 5 out of 6 MDs included in Figure 4 and only 7 out of 14 CMDs?

We thank the reviewer for their questions, and we will reply to them separately below.

Question 1: Re. Figures 2 - 4: In Figure 2, I question whether the ordering of traits (among MDs and CMDs, respectively) by heritability is helpful. Perhaps some order based on the nature of these traits, and how similar/related they are to one another, would facilitate understanding. E.g., list next to on both forms of ICA (unruptured and all). After all, the estimated heritabilities per se are not an organizing pillar of this paper. And whatever order is chosen in Figure 2 should be maintained in Figure 3 and 4 for consistency.

We thank the reviewer for this comment. We opted to retain the h^2 order of Figure 2. We investigated a more evidence-based clustering going beyond the mere separation of MD and CMD, but these often resulted in more arbitrary decisions than performing a magnitude-based ordering. We did take the reviewers comment to keep the order of figure 2 and maintained it in Figure 3 and 4.

Question 2: Also, why are only 5 out of 6 MDs included in Figure 4 and only 7 out of 14 CMDs?

To perform the analyses presented in figure 4 we need multiple components: 1.) prevalence of the disorders, 2.) the heritability of both disorders, 3.) the genetic correlation between the 2 disorders, and 4.) a measure of phenotypic comorbidity between the 2 disorders (in this case we used hazard ratios). We can derive point 1 to 3 ourselves, however, point 4 needs to be extract from a large epidemiological study. In this study we used measures of comorbidity presented by Moment at al, 2020 (the largest study to date using Danish register data) which overlapped with 5 out of 6 MDs and 7 of our 14 CMDs.

We stated this in the results section:

“As a measures of comorbidity, we used hazard ratios previously determined in the Danish population¹, allowing us to estimate the relative heritable and environmental components for 35 of the 90 pairs of MD-CVD.”

Reviewer #2 (Remarks to the Author):

Thank you for the opportunity to review the manuscript. Dr Meijssen et al investigated the genetic and environmental contributions underlying the observed comorbidity between six MDs and 14 CMDs and reported that genetic factors contributed about 50% to the comorbidity of schizophrenia, affective disorders, and autism spectrum disorder with CMDs, whereas the comorbidity of attention-deficit/hyperactivity disorder and anorexia with CMDs was mainly or fully driven by environmental factors. This work shed light on preventing and treating CMD in individuals with mental disorders. The statistical methods were corrected and the manuscript was generally well-written. I only have one concern. The authors reported that the age at first diagnosis for CVD was very young, for example, 1 year old? Please recheck the data and perform more sensitivity analyses if necessary.

We thank the reviewer for these kind words and for asking this question:

“The authors reported that the age at first diagnosis for CVD was very young, for example, 1 year old? Please recheck the data and perform more sensitivity analyses if necessary.”

We checked the age of first diagnosis for individuals diagnosed at different ages up till age 40 in the full Danish register and in the narrow window. Here we can show that a small number of individuals will get a diagnosis relatively early in life.

CVD	Entire register				1981-2005			
	N _{AOD} ≤10	N _{AOD} ≤20	N _{AOD} ≤30	N _{AOD} ≤40	N _{AOD} ≤10	N _{AOD} ≤20	N _{AOD} ≤30	N _{AOD} ≤40
OBE	10195	26302	79820	138584	7008	20248	52232	57325
T2D	1426	5483	13400	28477	258	976	2563	2976
HYP	355	1903	8803	30862	191	1443	4211	5019
AP	63	430	2041	11376	56	354	918	1084
CAD	84	357	1816	11346	65	225	608	736
CIHD	45	179	782	5364	35	111	280	356
AF	79	610	2648	7738	48	435	1199	1390
HF	255	516	1139	3220	178	396	692	771
AS	863	2231	5812	14993	523	1157	2037	2259
AIIS	408	934	2784	8518	254	599	1258	1442
ICA (R&U)	132	771	2575	5967	79	371	762	816
PAD	464	1953	5101	12249	223	1176	2040	2238
VET	225	2345	9914	22971	113	1343	3730	4191
SAH	120	714	2400	630	70	321	652	696
ICA (U)	14	65	213	5511	11	57	121	134

AOD = Age of first Diagnosis

For your convenience we have calculated the h² of hypertension and T2D after removing individuals diagnosed before age 20. Furthermore, we used the same cut-off to calculate the genetic correlation between 2 of the biggest phenotypes as a sensitivity analysis. We observed no difference in the genetic correlation between Hypertension and MDD ($r_g=0.15$, 95% CI 0.1-0.21) and T2D and MDD ($r_g=0.22$, 95% CI 0.16-0.27) compared with previous Danish estimates that contained people diagnosed with a CVD after 1 year. Therefore, we do not think that our reported estimates will change substantially after removing individuals diagnosed before a specific age.

Reviewer #3 (Remarks to the Author):

The authors should be commended for this extensive body of work that seeks to advance understanding of mental disorders and their link with cardiometabolic diseases.

We thank the reviewer for these very kind words.

Major comments

Ancestry is a major factor to consider when conducting any genetic analyses. There is no mention of the ancestry of the population data. One could assume that all individuals are European ancestry (as nothing to the contrary is stated), however that is problematic, particularly given recent history of population movements (I am more aware of these for Sweden than Denmark admittedly). Has the genetic data been used to confirm the ancestry of the populations? More information about this is required (including justification if this has not been attempted).

We thank the reviewer for this in-depth comment. This topic has been greatly on our mind, and we discussed it to some extent in the discussion. To address your comments, we'd like to mention that to the best of our knowledge ancestry is not recorded in any of the registers in either Denmark or Sweden. Furthermore, there is no genetic data available of the whole population to infer ancestry outside of the iPSYCH cohort (which were filtered to be of white-Danish ancestry when performing most GWAS analyses). In the register data we did remove individuals who were born outside of Denmark and Sweden. We want to point out that this was not done for any ancestry-related reasons but to minimise the effect of left-handed censoring as these individuals may have been diagnosed in another country and we do not have access to that information. This means that we excluded individuals born abroad to Danish/Swedish citizens as well as individuals migrating to Denmark and Sweden. Both Denmark and Sweden are predominantly white with relatively recent large migration patterns from non-European countries therefore we assume that we extracted most individuals of European ancestry and indirectly removed individuals of non-European ancestry when filtering on country of birth.

Responds in manuscript:

We wrote in the Integrative Psychiatric Research Consortium 2012 (iPSYCH2012) cohort section:
"Finally, we extracted unrelated individuals of European ancestry leaving 77,082 individuals."

In the Case definition section, we added:

"To minimise the effect of left-handed censoring we removed individuals born outside of Denmark and Sweden as these individuals may have been diagnosed in another country. By doing so we excluded both Danish/Swedish citizens as well as individuals migrating to Denmark and Sweden. No information is recorded regarding terms such as "race", ancestry," or "ethnicity". However, both Denmark and Sweden are predominantly of white-European ancestry with relatively recent large migration patterns from non-European countries therefore we assume that we extracted mostly individuals of white-European ancestry and indirectly removed individuals of non-European ancestry when filtering on country of birth."

What is the rationale for considering depression and bipolar combined (as affective disorders) and then bipolar separately but not depression separately? I can see the logic for the combined affective disorder, given that the diagnoses are overlapping and that there can be some misdiagnoses (with unipolar depression being one end of the spectrum and bipolar the other end, with many clinical cases with presentation falling in the middle). But would it not be more logical to consider unipolar depression separately (as a more distinct phenotype) as well as the combined affective disorder? Or bipolar, depression and affective disorder? This is particularly interesting given the recent awareness of increasing prevalence of depression worldwide.

We thank the reviewer for this comment. However, given that this was also mentioned by reviewer 1 we copied our response to them again here.

We acknowledge that combining BP and MDD into AFF is less optimal and we thank the reviewer(s) for pointing this out. We have calculated the genetic correlations between MDD and the CMDs and report them here (Table below). We observed no significant difference with the AFF results mentioned throughout the paper. These results support our assumption that MDD backs up the bulk of AFF diagnoses therefore the results are highly similar.

	Denmark								Sweden			
	AFF				MDD				rg	se	l95	u95
	rg	se	l95	u95	rg	se	l95	u95				
Angina Pectoris	0.25	0.04	0.17	0.32	0.24	0.01	0.16	0.32	0.15	0.02	0.06	0.24
Any Ischemic Stroke	0.30	0.06	0.18	0.41	0.30	0.02	0.19	0.41	0.24	0.02	0.12	0.36
Any Stroke	0.30	0.06	0.18	0.43	0.31	0.02	0.19	0.44	0.27	0.02	0.14	0.40
Atrial Fibrillation	0.08	0.05	-0.01	0.17	0.08	0.01	0.00	0.17	0.08	0.02	-0.01	0.17
Chronic Ischemic Heart Disease	0.19	0.04	0.12	0.26	0.18	0.01	0.10	0.26	0.12	0.02	0.04	0.21
Coronary Artery Disease	0.23	0.04	0.15	0.30	0.22	0.01	0.15	0.30	0.17	0.02	0.04	0.29
Heart Failure	0.24	0.06	0.12	0.36	0.25	0.02	0.13	0.37	0.21	0.02	0.11	0.31
Hypertension	0.16	0.03	0.11	0.21	0.16	0.01	0.10	0.21	0.16	0.01	0.10	0.21
Intracranial Aneurysms (Ruptured and Unruptured)	0.18	0.07	0.05	0.31	0.18	0.03	0.03	0.34	0.23	0.02	0.09	0.37
Intracranial Aneurysms (Unruptured)	0.14	0.11	-0.08	0.36	0.23	0.08	-0.05	0.53	0.08	0.03	-0.06	0.23
Peripheral Artery Disease	0.27	0.08	0.11	0.43	0.27	0.02	0.11	0.42	0.22	0.01	0.13	0.32
Subarachnoid Hemorrhage	0.20	0.08	0.03	0.36	0.15	0.02	0.00	0.31	0.32	0.02	0.12	0.52
Type-2 diabetes	0.20	0.04	0.13	0.27	0.20	0.01	0.13	0.28	0.22	0.01	0.16	0.27
Venous Thromboembolism	0.24	0.05	0.17	0.32	0.24	0.02	0.14	0.34	0.15	0.01	0.06	0.23
Obesity	0.48	0.02	0.36	0.60	0.48	0.02	0.35	0.60	0.15	0.02	0.06	0.24

We added this statement to the discussion and added the results in supplementary table S4B
"We calculated the rg between MDD and all CMD as observed no significant difference with the AFF results supporting our assumption that MDD makes up the majority of AFF diagnoses (Table S4B)."

A similar point can be made considering obesity, as a conspicuously missing cardiometabolic phenotype. What is the rationale for not including this, as it is perhaps the most obvious (from observational studies as well as biological plausibility) link between mental disorders and the subsequent CMD investigated? If there is good rationale, this should be made clear.

Again, we thank the reviewer for this question, which was also raised by reviewer 1. Here we provide the same answer we provided to reviewer 1.

We fully agree that obesity should have been studied in this large body of work; thank you for bringing this up. Due to a coding mistake, we were under the wrong impression that obesity was poorly coded in the Danish registers hence we excluded this phenotype from the start. We now have identified over 350K diagnosed individuals with obesity in the registers and included register based genetic estimates into the paper (Figure 2 and 3 and supplementary table S2 and S4) and added the results here as well for your convenience.

Genetic correlation between obesity and mental disorders												
	Denmark				Sweden				Meta-analysis			
	rg	se	l95	u95	rg	se	l95	u95	rg	se	l95	u95
ADHD	0.43	0.02	0.35	0.50	0.51	0.02	0.41	0.62	0.47	0.04	0.38	0.55
ASD	0.11	0.02	0.01	0.21	0.31	0.02	0.23	0.40	0.52	0.04	0.43	0.60
ANO	-0.23	0.03	-0.39	-0.07	-0.25	0.03	-0.41	-0.10	-0.24	0.02	-0.29	-0.20
AFF	0.48	0.02	0.36	0.60	0.56	0.03	0.41	0.71	0.21	0.10	0.01	0.41
BIP	0.32	0.04	0.13	0.50	0.30	0.03	0.18	0.42	0.31	0.02	0.26	0.35
SCZ	0.27	0.03	0.14	0.39	0.08	0.06	-0.17	0.32	0.18	0.10	-0.02	0.38

Regarding limitations, to what extent does the use of electronic records from hospital treatment influence the results (compared to say general practice diagnoses)?

This is indeed a very important question and one we haven't addressed enough in the paper, we feel. We thank the reviewer for bringing this to our attention. In our opinion, this strongly depends on the disorder being researched. Regarding the mental disorders, in previous work (Pasman, Meijsen, Haram et al, 2023) we have shown that there is a large discrepancy between the number of hospital-treated MDD cases in Denmark and Sweden. We concluded that in Denmark MDD is treated more on the GP (family doctor) level and only severe cases are referred to the hospital for specialist care, making the hospital treated cases more "extreme samples". We did not find any evidence of this in Sweden leaving us to believe that MDD is more likely to be treated at the hospital level than the GP level. However, we think it is unlikely that other mental disorders such a schizophrenia, bipolar disorders, ADHD, anorexia, and autism are treated on the GP level. Regarding cardiometabolic disease, we hypothesis that obesity, hypertension, angina, and type-2 diabetes have the potential to be more extreme cases as they are, in a less severe form, also often treated on the local GP level, however other CMDs tend to be more acute and treated in hospitals exclusively. Especially obesity is treated on a local level and individuals diagnosed with obesity at the hospital often receive this as a secondary diagnosis after entering the hospital for another disorder.

We added this section to the discussion

"Fourth, the use of hospital records may capture more extreme phenotype as compared to medical records from family doctors and general practitioners, however we hypothesise that this is both country and disorder specific."

The limitations suggest that the results might not translate well to “non-Scandinavian” countries, hinting that within Scandinavia, perhaps the environmental/genetic factors might be similar across other Scandinavian countries. Is this is fair assumption?

We'd like to thank the reviewer for this comment. We do believe that this is a fair assumption, however there are some important caveats that we should have addressed more clearly. First, we do not believe that this relates necessarily to the genetic similarity between Scandinavian countries but more to the fact that Scandinavia countries share highly similar health care systems and health information recording systems. Countries inside and outside Europe operate different medical systems and often also document similar disorders differently for a wide variety of reasons. While our cumulative incidence-based method accounts for within-country differences to some extent, we cannot rule out that country-specific health practices as well as recording practices would result in different estimates. We think that the estimates we present should not be qualitatively different for most countries with universal health systems.

Minor comments

With the methods being presented after the results, please check that abbreviations are defined at first use (SNP, CMD)

We'd like to thank the reviewer for spotting these mistakes and we believe we have now addressed these throughout the paper.

Consistency should be improved: population-complete genealogies or near-complete? Cardiovascular and metabolic or cardiometabolic?

We agree that “population-complete genealogies” may be wrong therefore we replaced all of these to “near-complete genealogies”. We also now consistently refer to CMDs as “cardiometabolic”. Both have been modified throughout the paper.

Please define abbreviations in supplementary table footnotes

To the best of our knowledge, we have done this throughout the paper. For example, Supplementary table S6 below the table has this text:

“ADHD = attention deficit/hyperactivity disorder, ASD = Autism spectrum disorder, ANO = Anorexia nervosa, AFF = Affective disorder, BIP = Bipolar disorder, SCZ = Schizophrenia, ICA (R&U) = intracranial aneurysms (ruptured and unruptured), ICA (U) = intracranial aneurysms (unruptured), SAH = Subarachnoid hemorrhage, PAD = Peripheral artery disease, VET = Venous thromboembolism, AS = Any stroke, AIS = Any ischemic stroke, AF = Atrial fibrillation, HF = Heart failure, CAD = Coronary artery disease, T2D = Type-2 diabetes, AP = Angina pectoris, CIHD = Chronic ischemic heart disease, HYP = Hypertension, EUR = European and EAS = East-Asian.”

We hope that this cleared up any potential confusion.

Please check for typos: line 156, negligible? Line 263, recoding, or recording?

Thank you for spotting this and we have now adjusted it.

REVIEWERS' COMMENTS

Reviewer #1 (Remarks to the Author):

The authors have spent much effort in addressing every single comment by the reviewers. They have generated additional data and/or added clarifying text and information and changed and altered various statements and terminology. And even in situations where the authors have not taken a reviewer's suggested route, they have provided the reasons for their decision, making at the very least a reasonably persuasive case that their choice is not inappropriate or worse than alternative options.

I feel that these changes have improved the quality of the manuscript. I can see even more clearly now that this manuscript is quite different from SNP-based studies that estimate and report correlations between mental and cardiometabolic diseases and deserves to be published. I have no more concerns.

Reviewer #2 (Remarks to the Author):

The authors have addressed all my concerns. Thanks!

Reviewer #3 (Remarks to the Author):

All of my concerns have been addressed in a satisfactory manner.